# OpenReview forum: "RedTeamCUA: Realistic Adversarial Testing of Computer-Use Agents in Hybrid Web-OS Environments"
_ICLR.cc/2026/Conference — ICLR 2026 Oral_

### Official Review · Reviewer_Djvv · 2025-10-27

**Soundness:** 4
**Presentation:** 4
**Contribution:** 4
**Rating:** 6
**Confidence:** 5

**Summary:**

The paper introduces RedTeamCUA, a novel adversarial testing framework designed to evaluate the susceptibility of Computer-Use Agents (CUAs) to prompt injection in realistic, controlled, hybrid environments spanning both the web and the operating system. The core of the framework integrates a VM-based OS environment (based on OSWorld) with Docker-based containers. This design allows for controlled, end-to-end evaluation of attacks that cross the web and OS boundary, such as an injection on a forum leading to a harmful local OS action.

The authors also propose RTC-BENCH, a comprehensive benchmark of 864 examples that investigates realistic hybrid attack scenarios targeting fundamental security violations categorized by the Confidentiality, Integrity, and Availability (CIA) triad. The framework also introduces a Decoupled Eval setting, which bypasses CUA navigation limits to isolate and focus on core adversarial robustness.

Benchmarking frontier CUAs, including Claude and Operator revealed significant vulnerabilities. Even the most secure agent, Operator, exhibited an Attack Success Rate (ASR) of 7.6% in the Decoupled setting, while Claude 3.7 Sonnet reached an ASR of 42.9% and in realistic End2End settings, ASRs reached ~50%. The authors conclude that CUA threats are already tangible and current defenses, including both built-in CUA mechanisms and four evaluated defense methods, are insufficient.

**Strengths:**

1. The first controlled, integrated framework for adversarial testing across both realistic OS and web environments.

2. Decoupled Evaluation Setting effectively isolates an agent’s true adversarial robustness from confounding factors like navigational capability, providing a clear measure of vulnerability once an injection is encountered.

3. The empirical results demonstrate that frontier CUAs are highly vulnerable, with Attack Success Rates reaching up to 50% in end-to-end settings, confirming that these are immediate, tangible risks.

4. The threat model is more realistic than much prior work.

**Weaknesses:**

1. The current findings are primarily limited to a few closed source models and do not test on open source CUA models like UI-TARS 1.5 70B.

2. The realistic End2End evaluation was performed only on a subset of 50 tasks out of the 864 examples in the benchmark.

**Questions:**

N/A.

---

> ### Author Response · Authors · 2025-11-21
> **Responses**
>
> We thank the reviewer for recognizing the novelty of our framework, the utility of the decoupled eval setting, the realistic threat model, and the significance of our findings.
>
>
> **[Open-Source CUAs (W1)]**
>
> We have included additional analysis with a newly released agent based on Qwen-3-VL to expand on the scope of our analysis. Please refer to the general response for more details on the omission of alternative open-source CUAs options and additional results for Qwen-3-VL.
>
>
>
> **[End2End Evaluation (W2)]**
>
> We agree that expanding the End2End evaluation would further strengthen our analysis, as this setting most closely mirrors realistic deployment conditions. However, it incurs substantial computational costs, e.g., reproducing the End2End results in Table 2 alone required approximately 350 USD in total (not to mention substantial costs in other parts of the experiments; the total cost is around 8,000 USD).
>
> Given the cost constraints, in our paper, we carefully selected a subset of 50 tasks resulting in the highest ASR under the Decoupled Eval setting. These tasks capture the most vulnerable scenarios, enabling us to reveal key adversarial risks while remaining in budget. Furthermore, adversarial risk fundamentally concerns worst-case behavior: even a single successful malicious manipulation indicates that the agent can be misled and therefore requires mitigation. In this light, evaluating these most vulnerable cases is sufficient to support our claim that current CUAs can already cause tangible harm, underscoring the urgency of addressing these risks.
>
> We will discuss these aspects more clearly in the revision and consider broadening the End2End evaluation in the future as resources permit.

---

### Official Review · Reviewer_aku4 · 2025-10-29

**Soundness:** 3
**Presentation:** 3
**Contribution:** 2
**Rating:** 6
**Confidence:** 2

**Summary:**

The authors propose RedTeamCUA, a benchmark for "hybrid web-OS" indirect prompt injections on a number of frontier models / agents. They find that most tested agent+scenario+model combinations are vulnerable to attacks in their benchmark in both targeted and end-to-end experiments. In addition, a suite of recent defenses are circumvented.

**Strengths:**

- Well written and designed benchmark.
- Addresses an important niche of attacks on CUA and hybrid models.
- Interesting findings on the limitations of current defense frameworks in this regime.

**Weaknesses:**

- Novelty: The method seems to build on top of and combine existing benchmarks and attacks. In a future version, I would like to see the authors spell how their benchmark differs a little more.
- Attacks: Only one format of attack was evaluate. Would be nice to see evaluation of more.
- Sanitization: How sensitive are results to realistic UI noise (extra messages/files)?

**Questions:**

- How sensitive are results to realistic UI noise (extra messages/files)?
- Any signal as to whether results transfer to other common platforms (e.g., wikis, issue trackers)?
- In End2End, how much of the ASR drop is due to navigation vs perception vs tool-use? A per-failure taxonomy would be very interesting.

---

> ### Author Response · Authors · 2025-11-21
> **Responses (1/2)**
>
> We thank the reviewer for acknowledging our well-designed benchmark, the significance of our work, as well as the interesting findings on the limitations of current defense frameworks.
>
> **[Novelty of Our Sandbox & Benchmark (W1)]**
>
> We appreciate the reviewer's feedback and clarify the novelty of both our sandbox and our benchmark below.
>
> - **Sandbox:**
>
> Our sandbox environment builds upon three established frameworks: OSWorld to provide a VM-based OS, and both WebArena and TheAgentCompany to provide Docker-based web replicas. However, none of these frameworks are directly suitable for adversarial risk assessment: OSWorld permits unrestricted browser access to live webpages, introducing real-world safety concerns during testing, while WebArena and TheAgentCompany exclusively support web browsers, preventing studies of OS-related adversarial harms. Motivated by the goal of reliably and comprehensively assessing CUA risks across both the web and OS, we carefully integrate the strengths of these frameworks to build a hybrid sandbox that supports adversarial evaluations across both the web and OS.  We additionally introduce substantial code changes implementing core features tailored specifically for adversarial evaluation (detailed in lines 215–230). These include platform-specific injection scripts, extensions to OSWorld’s configuration files to enable flexible and controlled injection, and a Decoupled Evaluation setting using pre-processed actions to place a CUA directly at the site of injection for focused adversarial analysis. Additional discussion on the limitations of prior frameworks and a detailed comparison of our work and prior work are provided in Section 2.2, Appendix D, and Table 5.
>
> - **Benchmark:**
>
> In this work, **we newly introduce RTC-Bench**, a comprehensive benchmark featuring 864 indirect prompt injection examples designed to explore hybrid web-OS attack pathways that are only enabled by our sandbox.
>
> The benchmark first introduces a novel benign goal formulation, featuring 9 benign task scenarios where CUAs assist users in retrieving web knowledge for local OS execution. We then introduce 24 adversarial goals grounded in violations of fundamental CIA (Confidentiality, Integrity, Availability) security principles. Our adversarial setup emphasizes web-based adversarial injection that results in harmful outcomes to the user's local system, exploring novel Web → OS and Web → OS → Web attacks not previously possible in prior adversarial frameworks. RTC-Bench also introduces a rigorous threat model that includes realistic attacker constraints and a contextualized injection strategy. Additional discussion of RTC-Bench is provided in Section 4.
>
> We will emphasize these distinctions more explicitly in the next revision, as suggested by the reviewer.

---

> > ### Author Response · Authors · 2025-11-21
> > **Responses (2/2)**
> >
> > **[Attack Formats (W2)]**
> >
> > Regarding “Only one format of attack”, let us first clarify: our attack has more than one format for the same benign goal. Particularly, the injected content can be instantiated via either code or natural language (line 280), representing two classical ways of injection. Specific examples can be found in Appendix B.4, and results on two different instantiations are presented in Appendix I.1.
> >
> > Moreover, each injection (attack) is customized according to different benign goals and platforms, ensuring that the injected adversarial content aligns with the environmental semantics, making it more realistic and effective in practice. Therefore, we indeed have more than one attack format.
> >
> > Additionally, in our preliminary experiments, we also explored several alternative attack strategies and formats, including non-contextualized injections, variants without commonly used “important message” cues, and versions lacking fake, environment-specific rationales. However, these alternatives consistently exhibited lower attack success rates across multiple CUAs. Based on these careful explorations, we deliberately selected our final injection strategy to maximize both realism and efficacy.
> >
> > We would also like to emphasize that the primary goal of our work is not to exhaustively enumerate a wide variety of attack formats or attack strategies, but RedTeamCUA provides a hybrid sandbox to study cross-platform attacks in realistic conditions. It can also readily support various threat models and different attack vectors, including multimodal pop-up injection. Built upon this flexible sandbox, our benchmark, the effective strategy we discovered, and experimental findings altogether highlight critical vulnerabilities in current CUA designs and underscore the urgent need for more robust CUAs.
> >
> > **[Sensitivity to UI Noise? (Q1)]**
> >
> > In principle, a sufficiently capable CUA with robust instruction-following abilities should maintain consistent performance despite extraneous UI elements. However, given the fact that LLM often suffer from context rot issues, where the accumulation of irrelevant information dilutes the model’s attention and thus leads to drifting behaviors, the LLM-based CUA might face similar issues as well.
> >
> > **[Transfer to Other Platforms? (Q2)]**
> >
> > Whether the results transfer to other common platforms will depend on the specific platforms and the specific adversarial goal. Even when employing similar attack strategies, identical adversarial goals can yield substantially different outcomes across platforms, as demonstrated in Table 1 and Figure 2.
> >
> >
> > That said, wikis share key characteristics with the ownCloud platform studied in our work: both organize information in a document-like structure and host many documents. Similarly, issue trackers, such as GitHub Issues, are similar to the Reddit platform included in our study, where online users leave messages under a certain topic. Therefore, our findings are likely to transfer, at least partially, to these two types of platforms.
> >
> >
> >
> > **[Failure Taxonomy (Q3)]**
> >
> > Thanks for the great suggestion. Below is the failure taxonomy:
> >
> > | Model              | Triggered Check  | Failed Navigation | Ignored Injection | Limited Capability (Adv. Goal) | Successful Attack |
> > |--------------------|----------------------------------|-------------------|-------------------|-------------------------------|-------------------|
> > | Operator           | 33                               | 4                 | 6                 | 2                             | 5                 |
> > | Operator w/o check | 0                                | 11                | 16                | 2                             | 21                |
> > | Claude 3.7         | 0                                | 15                | 3                 | 7                             | 25                |
> > | Claude 4.0         | 0                                | 11                | 9                 | 6                             | 24                |
> >
> > We categorize failure cases into the following types:
> >
> > - **Failed Navigation**: The agent is unable to navigate to the target page required for the task.
> > - **Ignored Injection**: The agent reaches the target page but completely ignores the injected adversarial content.
> > - **Limited Capability (Adv. Goal)**: The injection successfully misleads the agent, but the agent lacks the capability to complete the adversarial goal after being misled.
> > - **Triggered Check** (Operator only): The built-in permission/safety check of Operator is triggered and requests human intervention, which we count as a failure of the attack.
> >
> > We will integrate it into the next revision to augment our analysis.

---

### Official Review · Reviewer_MWPP · 2025-10-31

**Soundness:** 4
**Presentation:** 4
**Contribution:** 3
**Rating:** 8
**Confidence:** 4

**Summary:**

Authors propose a new framework (redTeamCUA) + benchmark (RTC Bench) by combining OSWorld environment w/ WebArena and TheAgentCompany.

Combination of benign goals and adversarial goals similarly to AgentDojo:
(9 benign goals × 24 adversarial goals × 4 types of instantiation)
- instantiation means: Code vs. NL prompt injection x General vs. Specific benign goal

Interesting seeting: Decoupled Eval, where the agent is brought to the point of adversarial injection (hardcoded prior tool calls?)

Metrics are ASR and AR (attempt rate for adversarial goals).

They show high ASR for all evaluated models, except for Operator, which still has 7.6%.

**Strengths:**

- OSWorld backbone allows for hybrid attacks over both OS and web
- Realistic threat model
- Decoupled evaluation setting is good for helping weaker capability models reach the point of prompt injection, though it might be somewhat un-natural depending on how the tool-calling/traces are hard-coded to the agent history
- Great having both Web -> OS -> Web and Web -> OS adversarial scenarios

**Weaknesses:**

- The large number of 864 examples is only achieved by cross-product of benign, injection, and instantiation. The number of benign tasks (9) might be too small to accurately estimate agent utility, which is critical in any security benchmark (otherwise a useless agent might have perfect security).
- I disagree with the the assessment that the Doomarena threat model is requires full webpage control; even though the authors note that the banners and pop-up attacks are injected into the web page (e.g. of PostMill) by modifying the DOM, these elements are typically 3rd party content, which is editable without full page access by, e.g., submitting the attacks through some advertising platform.

**Questions:**

- You use the CIA taxonomy of threats. I'm curious where something like Direct Harm (e.g. send money to attacker) would land.

---

> ### Author Response · Authors · 2025-11-21
> **Responses**
>
> We would like to thank the reviewers for recognizing our realistic threat model, the benefits of our decoupled evaluation setting, and the hybrid attack pathways enabled by our hybrid sandbox.
>
> **[Agent Utility (W1)]**
>
> We totally agree with the reviewer on the importance of accurately estimating agent utility, and that “a useless agent might have perfect security”.  First, we hope to clarify that our assessment of agent utility is *NOT* entirely based on the number of benign tasks (9). Instead, we carefully selected existing CUAs for our experiments based on the following considerations (please also see our general response):
> - We restrict our analysis to CUAs that are reasonably capable to avoid spurious security conclusions. We specifically focus on CUAs with strong OSWorld performance to ensure sufficient capabilities; for example, Claude Sonnet 3.5 and 3.7 ranked as the top two CUAs with max steps set to 50 at the time of submission. In addition, we perform additional sanity checks to ensure that agents can properly follow instructions, navigate, and interact when no attack is present. We encourage all users of our benchmark to follow this practice to ensure reliable evaluation.
>
>
> Moreover, we implement several measures to separate true CUA security from spurious security due to limited capability. Within our study, we take the following steps to disambiguate agent robustness to indirect prompt injection from capability limitations:
>
>
> - **Decoupled Evaluation**: We note the impacts of navigational limitations on adversarial analysis in lines 223-227, highlighting that an inability to reach injection for models such as GPT-4o does not imply robustness to the injection once encountered. To account for this limitation, we introduce the *Decoupled Evaluation* setting as a core feature of RedTeamCUA, using preprocessed actions to place the target CUA at the injection point during task setup. Through this, we further mitigate spurious security conclusions by isolating adversarial robustness from any navigational limitations.
>
>
> - **Attempt Rate**: As discussed in lines 366-369, adversarial analysis may be impacted by instances where CUAs are misled by an injection but fail to fully complete the adversarial goal due to capability constraints. To address this, we include an *Attempt Rate* evaluation metric, using a fine-grained LLM-as-a-Judge approach to assess whether the evaluated CUA attempted the adversarial goal at any point in the trajectory regardless of full completion. This allows us to further capture vulnerabilities that might otherwise be masked by limited utility.
>
>
> **[DoomArena’s Threat Model (W2)]**
>
> We appreciate the clarification regarding the DoomArena’s threat model. We acknowledge that certain web elements such as pop-ups, banners, or ads may be injected through third-party channels without technically requiring full webpage control.
>
>
> We will further clarify distinctions related to DoomArena’s threat model in the revised manuscript. We treat this as a minor issue that is easy to address and please feel free to let us know if there are further questions.
>
> **[CIA Taxonomy and Direct Harm (Q1)]**
>
> The CIA taxonomy is used in our work to classify various harms based on information security, with *Confidentiality* referring to unauthorized disclosure of sensitive information, *Integrity* referring to negative impacts on the trustworthiness and accuracy of data, and *Availability* referring to a lack of reliable access to data and systems. Given this, tasks that result in “direct harm” to the user such as “sending money to attacker” would be classified as an Integrity violation because it involves the unauthorized manipulation of the data state (i.e., the user’s balance).

---

### Official Review · Reviewer_y1z7 · 2025-11-01

**Soundness:** 2
**Presentation:** 2
**Contribution:** 2
**Rating:** 4
**Confidence:** 4

**Summary:**

The paper presents REDTEAMCUA, a framework for realistic adversarial testing of computer-use agents (CUAs) across hybrid web–OS environments. It builds a VM + Docker sandbox and an RTC-BENCH benchmark with 864 adversarial cases to evaluate indirect prompt-injection risks. Experiments show serious vulnerabilities in leading CUAs (e.g., ASR up to 66%, AR > 90%), even for models like Claude 3.7 Sonnet | CUA and Operator. Existing defenses such as LlamaFirewall and Meta SecAlign offer limited protection, revealing urgent needs for stronger security in CUAs.

**Strengths:**

– Proposes a well-designed, hybrid sandbox integrating web and OS layers, bridging realism and safety in adversarial testing.

– Builds a large-scale, systematic benchmark (RTC-BENCH) grounded in realistic tasks and security principles (CIA triad).

– Provides comprehensive empirical results with both execution-based and LLM-judge metrics, revealing concrete weaknesses in current frontier CUAs.

– Conducts thoughtful analysis comparing adapted LLM agents vs. specialized CUAs, and offers valuable insight into the trade-off between autonomy and safety.

**Weaknesses:**

– some closed-source CUAs evaluated (GPT-4o, Claude 3.5/3.7 Sonnet, Claude 4 Opus, Operator) dominate the study; no strong open-source CUAs (e.g., UI-TARS 2, OpenCUA) are included, limiting reproducibility and community relevance. More closed-source CUAs and open-source CUAs need to be included.

– The defense evaluation is superficial—existing methods are merely tested rather than extended or improved.

– Provides limited mechanistic analysis of why specific CUAs succumb to injection (e.g., reasoning path, memory, or grounding failure).

– Some sections are overly descriptive and lengthy, diluting the main technical insights.

**Questions:**

see weakness.

---

> ### Author Response · Authors · 2025-11-21
> **Responses (1/2)**
>
> We thank the reviewer for acknowledging our well-designed sandbox, systematic benchmark, and concrete analysis of frontier CUAs grounded in realistic tasks, environments, and security principles.
>
> **[Agent Selection (W1)]**
>
> We have included additional analysis with Claude 4.5 Sonnet and Qwen-3-VL to expand on the scope of our analysis. Please refer to the general response for more details on our study’s comprehensive CUA selection, the omission of various open-source CUAs, and additional results for both Claude 4.5 Sonnet and Qwen-3-VL.
>
> **[Defense Extension and Improvement (W2)]**
>
> Please note that the primary goal of our study is to provide a comprehensive risk assessment that uncovers vulnerabilities in existing CUAs and existing defense strategies under realistic, secure conditions. The systematic analysis we present offers key insights into the current state of CUA security, which can better inform and guide improvements to defense in future work. As shown in the analysis discussed below, the development of effective defense methods for prompt injection is a significant challenge that warrants dedicated research studies (similar to  AGrail [1], CaMel [2] where novel defense methods are developed).
>
> To perform comprehensive analysis of defense strategies, we examined two categories of prompt injection defenses: (1) **System-Level Defenses**, which prevent exposure to injection through additional detectors, monitors, or prompting; and (2) **Model-Level Defenses**, which build in security by training a foundation model to prioritize trusted instructions over untrusted data. Overall, our evaluation spans a diverse set of defenses used in current CUAs to enable a comprehensive risk assessment, including permission checks, external guardrails, defensive system prompts, and dedicated defense training.
>
> We first evaluated the built-in confirmation and safety checks for Operator, which reached the lowest ASR of ~7.6% by requesting explicit user permission prior to executing critical or high-risk actions. However, as shown in Table 1, we also demonstrated a substantial ASR of ~31% when evaluating an Operator (w/o checks) variant that reflects the absence of reliable human supervision. This highlights the current necessity for human oversight to enable sufficient CUA guardrails and the need for better security mechanisms for truly autonomous CUAs.
>
> We then explored four additional defense methods on 50 high-risk examples from RTC-Bench that achieved the highest ASR in our experiments, allowing us to assess worst-case behavior across the most vulnerable scenarios. We found the following results:
>
> **System-Level Defenses:**
> - **External Guardrails**: We evaluated the recent LlamaFirewall [3] and PromptArmor [4], finding even the best variant is only capable of detecting 30% of injections (Appendix H.1).
> - **Defensive System Prompt**: We developed a custom defensive system prompt that instructs the agent to detect injections and prioritize the original instruction. As shown in Appendix H.2,  this approach remains insufficient, with ASR for Claude 3.7 Sonnet | CUA and Operator (w/o checks) still approaching 50% in End2End evaluation.
>
> **Model-Level Defenses:**
> - **Foundation Models**: We evaluated Meta SecAlign [5], an open-source foundation model trained with DPO specifically to resist prompt injection by prioritizing user instructions over untrusted environment inputs. Our results in Appendix H.3 demonstrate that Meta SecAlign still reaches an Attempt Rate of 52% for malicious instructions, indicating that it fails to recognize half of the tested injections despite dedicated training.
>
> Through this, we identify notable limitations for existing defenses and urge the community to develop effective defense strategies to support safe deployment of CUAs in future work.
>
> [1] Luo, Weidi et al. “AGrail: A Lifelong Agent Guardrail with Effective and Adaptive Safety Detection.” Annual Meeting of the Association for Computational Linguistics (2025).
>
> [2] Debenedetti, Edoardo, et al. "Defeating prompt injections by design." arXiv preprint arXiv:2503.18813 (2025).
>
> [3] Chennabasappa, Sahana, et al. "Llamafirewall: An open source guardrail system for building secure ai agents." arXiv preprint arXiv:2505.03574 (2025).
>
> [4] Shi, Tianneng, et al. "Promptarmor: Simple yet effective prompt injection defenses." arXiv preprint arXiv:2507.15219 (2025).
>
> [5] Chen, Sizhe, et al. "Meta SecAlign: A Secure Foundation LLM Against Prompt Injection Attacks." arXiv preprint arXiv:2507.02735 (2025).

---

> ### Author Response · Authors · 2025-11-21
> **Responses (2/2)**
>
> **[Mechanistic Analysis (W3)]**
>
> We agree that mechanistic analysis could provide valuable insight into which components of a CUA contribute most to vulnerabilities to indirect prompt injection. However, as discussed in our general response, conducting a meaningful and reliable adversarial risk assessment requires the inclusion of sufficiently capable CUAs, creating an inherent reliance on closed-source CUAs. The black-box nature of frontier CUAs from OpenAI and Anthropic, including a lack of transparency regarding internal components and access to underlying model weights, leaves us incapable of performing the mechanistic analysis needed to *definitively* explain why CUAs succumb to prompt injection.
>
> With that said, we agree that understanding the underlying causes of indirect prompt injection in frontier CUAs is essential. Given this, our analysis provides deep insights into what factors contribute to attack success:
>
>
> In lines 282-290, we note our threat model’s usage of adversarial instructions that are carefully contextualized to both the benign request and environment context. This is achieved by providing a deceptive rationale designed to appear consistent with plausible tasks for the given environment context (e.g., a CUA being tasked with installing termcolor for a Forum page related to the package, Fig. 1). We identify reduced ASR for several alternative attacks, including non-contextualized injections and attacks lacking environment-specific rationales during preliminary tests, suggesting that *contextualization is a critical factor for attack viability on frontier CUAs*.  We also hypothesize in lines 407-411 that *the naturalness of an adversarial goal within a specific web platform* (e.g., sending confidential data being more plausible on a messaging platform like RocketChat) may help explain the platform-specific differences in ASR observed for the same adversarial goal, as shown in Table 1.
>
> **These observations suggest that susceptibility to indirect prompt injection stems from a failure in the CUA’s task-specific reasoning rather than memory or grounding**. Frontier CUAs struggle to reliably distinguish instructions required for benign task completion from injected instructions that appear contextually plausible for a given task and environment context. This interpretation is further supported by our Utility Under Attack results (Appendix I.4), where the CUA shows no difficulty in preserving the benign task in memory and fully completing it even after pursuing the adversarial goal.
>
> Altogether, we believe our empirical results and analysis provide valuable insights into the causes of CUA vulnerabilities to prompt injection. We will make related discussions more clear in the paper and encourage future work to explore more.
>
> **[Writing Adjustments (W4)]**
>
> Thank you for the feedback regarding the length of certain sections. We will continue refining the writing throughout the review process to ensure the technical insights are presented clearly and concisely. We treat this as a minor issue that is easy to address and please feel free to follow up if you have more specific suggestions.

---

> > ### Comment · Reviewer_y1z7 · 2025-11-25
> > **Authors have adressed most of my concerns; I raise my score to 6**
> >
> > Thank you for the detailed rebuttal. The authors have addressed most of my concerns with clarifications and additional analysis. After considering both the rebuttal and the perspectives raised by other reviewers, I raise my score to 6.

---

> > > ### Author Response · Authors · 2025-11-25
> > > **Thank You for Your Response**
> > >
> > > Thank you for reviewing our rebuttal and for your updated assessment. We're glad we were able to clarify the points you raised and we will continue to incorporate your thoughtful feedback going forward.

---

### Author Response · Authors · 2025-11-21
**General Responses (1/3)**

We thank all reviewers for their time and comments. Below, we present the rationale behind our selection of agents. We also include our newly added experiments on Claude 4.5 Sonnet [1], the top-performing CUA on the OSWorld leaderboard [2] at the time of our rebuttal, and on Qwen-3-VL-235B-A22B-Thinking [3], the top-performing open-source CUA on OSWorld that is currently accessible.

**[Agent Selection]**

Adversarial risk evaluation of computer-use agents (CUAs) represents a timely and high-impact direction. To conduct a meaningful and reliable adversarial risk assessment, we selected agents representative of the current state of CUAs while **maintaining the fundamental principle that evaluated CUAs must be reasonably capable**. Otherwise, *(1) an incapable agent would not be adopted in real-world settings, limiting relevance to the community and (2) an underperforming agent may appear spuriously safe simply because it fails to understand or even perceive adversarial instructions in the first place.*

Nevertheless, computer-use tasks are inherently complex, featuring long-horizon trajectories and highly idiosyncratic environments. Given this complexity and the goal of reliable assessment, we deliberately focus on the strongest available agents as indicated by OSWorld [4] performance, one of the most representative and widely used benchmarks for computer use evaluation. At the time of submission, CUAs developed by OpenAI and Anthropic (e.g., Claude series and Operator) represent the most capable and frontier CUAs. As such, this rationale justifies our choice to adopt them as the main baselines in our work. Additionally, we performed sanity checks to verify their basic utility and ensure the validity of our results.

Beyond general capabilities, the agents selected for our study provide a strong foundation for systematic analysis of CUA security given the following considerations:

- **Diverse Developers**: Our evaluated CUAs include CUAs built by multiple independent organizations, such as Anthropic’s Claude series and OpenAI’s Operator.
- **Agent Types**: Our evaluated CUAs span two categories, including (a) Adapted LLM-based Agents, which are general-purpose LLMs scaffolded for computer-use (e.g. GPT-4o, Claude 3.5/3.7 Sonnet) and (b) Specialized Computer-Use Agents, which are models purposefully designed for computer-use scenarios (e.g. Operator, Claude 3.5/3.7 Sonnet | CUA).
- **Defense Strategies**:  Diverse approaches to mitigating indirect prompt injection are covered by our selected CUAs, including dedicated RL-based training and external guardrails outlined in Section 4.2 of the Claude 3.7 Sonnet system card [5] and permission/safety checks that enable human oversight for Operator [6]. According to their reports, Claude 3.7 Sonnet's defense strategies prevented prompt injections in 88% of cases on their  evaluation set, while Operator reduced susceptibility from 62% without mitigations to 23%, demonstrating the perceived robustness of these methods. We hope to evaluate the effectiveness of these defense strategies under our adversarial attack.

Together, these choices ensure comprehensive coverage of the current CUA landscape (in terms of agent types, developers and defense strategies) and support a systematic assessment of CUA security vulnerabilities.

---

> ### Author Response · Authors · 2025-11-21
> **General Responses (2/3)**
>
> **[Open-Source Agent Omissions]**
>
> We also attempted to incorporate top-performing open-source agents from the OSWorld leaderboard, including those noted by the reviewers, at submission time to broaden our evaluation scope, but encountered several practical constraints.
>
> - **UI-TARS-1.5** [7] and **UI-TARS-2** [8] (as requested by the reviewer y1z7 and the reviewer Djvv) did not provide publicly accessible model weights.
> - **OpenCUA** [9] (as requested by the reviewer y1z7) released model weights but lacks support from high-throughput inference frameworks (e.g., vLLM, SGLang), making 50-step rollouts prohibitively time-consuming for both our evaluation and practical deployment.
> - **UI-TARS-1.5-7B**, a smaller variant of the UI-TARS-1.5 referenced above, was therefore the highest-performing open-source CUA on the OSWorld leaderboard that was publicly accessible at the time of submission. Unfortunately, it failed our basic sanity checks, struggling with basic instruction following, navigation, and interaction.
>
>
> For this reason, we exclusively included the more capable closed-source CUAs in our submission, as detailed in lines 355–360 of the paper. Nevertheless, we appreciate the suggestion to evaluate open-source CUAs in our work and provide additional analysis for a publicly accessible top-performing agent based on Qwen-3-VL (released after the submission deadline) in the section below.

---

> > ### Author Response · Authors · 2025-11-21
> > **General Responses (3/3)**
> >
> > **[Additional Analyses]**
> >
> > Between submission and review release, two notable CUA releases achieved strong OSWorld performance and were accessible for our analysis: (1) **Claude 4.5 Sonnet** [1], which demonstrated a substantial leap in computer-use ability to achieve the current SoTA on OSWorld; and (2) **Qwen3-VL-235B-A22B-Thinking** [3], an open-source model with 38.1% OSWorld accuracy, outperforming UI-TARS-1.5-7B in both our preliminary experiments and on OSWorld.  To strengthen our study, we further conducted additional end-to-end evaluations of both models, demonstrating the following results:
> >
> > **Claude 4.5 Sonnet**:
> > - Despite its capability improvements, Claude 4.5 Sonnet demonstrates an **End2End ASR of 60%**, the highest amongst all CUAs evaluated in our study.
> > - Paradoxically, the stronger instruction-following capabilities of Sonnet 4.5 improve performance but also make it more vulnerable to following malicious instructions embedded in the environment.
> > - This result underscores the urgent need for developing robust defenses alongside capability improvements to enable safe, real-world deployment.
> >
> >
> > **Qwen3-VL-235B-A22B-Thinking**:
> > - Qwen-3-VL successfully reached the target webpage (i.e., the webpage necessary for fulfilling the user goal and containing the injected malicious instruction) in 10 out of 50 tasks, indicating improved proficiency among open-source CUAs.
> > - Qwen-3VL achieves an overall End2End ASR of 8%.
> > - Notably, although its End2End ASR is lower than the safest agent (Operator) reported in our paper (10%), this does not necessarily mean that Qwen3-VL is safer. Rather, this is because it is less capable and often fails to reach the target webpage.
> >
> > We will continue to revise our manuscript to further explain the selection of CUAs within our study and include additional results from recent CUA releases that meet our capability criteria.
> >
> >
> > [1] Anthropic. Introducing Claude Sonnet 4.5. https://www.anthropic.com/news/claude-sonnet-4-5, 2025.
> >
> > [2] OSWorld Leaderboard. https://os-world.github.io/
> >
> > [3] Qwen AI. Qwen3-VL: Sharper Vision, Deeper Thought, Broader Action. https://qwen.ai/blog?id=99f0335c4ad9ff6153e517418d48535ab6d8afef&from=research.latest-advancements-list, 2025
> >
> > [4] Xie, Tianbao, et al. "Osworld: Benchmarking multimodal agents for open-ended tasks in real computer environments." Advances in Neural Information Processing Systems 37 (2024): 52040-52094.
> >
> > [5] Anthropic. Claude 3.7 sonnet system card, https://assets.anthropic.com/
> > m/785e231869ea8b3b/original/claude-3-7-sonnet-system-card.pdf, 2025
> >
> > [6] OpenAI. Computer-using agent, https://openai.com/index/computer-using-agent. 2025
> >
> > [7] ByteDance Seed. Ui-tars-1.5. https://seed-tars.com/1.5, 2025.
> >
> > [8] Wang, Haoming, et al. "Ui-tars-2 technical report: Advancing gui agent with multi-turn reinforcement learning." arXiv preprint arXiv:2509.02544 (2025).
> >
> > [9] Wang, Xinyuan, et al. "Opencua: Open foundations for computer-use agents." arXiv preprint arXiv:2508.09123 (2025).

---

### Author Response · Authors · 2025-11-25
**Follow-up on Author Response**

Dear reviewers,

We wanted to follow up and gently ask if you could take a moment to look over our author response. We really appreciate your time, and thank you again for all of the thoughtful feedback you've provided.

---

### Author Response · Authors · 2025-12-03
**Note to AC: Rebuttal Summary**

Dear Area Chair,

We sincerely appreciate you stepping in to oversee our submission given the current circumstances. We understand the substantial effort needed to evaluate our manuscript and prior discussion history, and seek to offer a summary of our rebuttal process to aid in your assessment.

We first thank all reviewers for their time and constructive comments. We appreciate their acknowledgement of (1) our well-designed, hybrid sandbox for adversarial testing across realistic OS and web environments (y1z7, MWPP, Djvv), including core features such as the Decoupled Evaluation setting to enable focused adversarial testing (MWPP, Djvv); (2) our large-scale, systematic benchmark RTC-Bench (y1z7, aku4) which incorporates a realistic threat model (MWPP, Djvv); and (3) our comprehensive empirical results revealing critical vulnerabilities in current frontier CUAs (y1z7, Djvv) and defense frameworks (aku4) across Decoupled and End2End evaluation settings.

We provide a detailed rebuttal to address critiques that have been raised by reviewers, emphasizing our comprehensive selection of agents for evaluation (y1z7, Djvv), the novelty of our benchmark and sandbox design (aku4), the depth of analysis of current defense strategies (y1z7), and our validation of agent utility via sanity checks and core features (MWPP). In particular, our general response further explains our rationale for agent selection, provides justification for our omission of several relevant open-source CUAs, and present new experiments evaluating vulnerabilities in Claude 4.5 Sonnet and Qwen-3-VL-235B-A22B-Thinking.

Through this rebuttal, we believe we resolved the concerns raised by reviewers and convinced reviewer y1z7 to raise their score. We hope this summary assists you in your final assessment.

---

### Meta-Review · Area_Chair_jsPT · 2026-01-06

**Summary:**

This paper introduces REDTEAMCUA, an adversarial testing framework for Computer-Use Agents featuring a hybrid sandbox that integrates VM-based OS environments with Docker-based web platforms. While existing frameworks either lack isolated web environments or cannot evaluate OS-level harms, this integration enables controlled evaluation of cross-environment attack scenarios where web-based injections trigger harmful OS actions. A key methodological contribution is the Decoupled Eval setting, which uses pre-processed actions to place agents directly at injection points, isolating adversarial robustness from navigational capability limitations. The authors also construct RTC-BENCH, a benchmark of 864 adversarial examples organized by the CIA security triad, featuring realistic threat models where attackers can only modify editable web content rather than assuming full webpage control. Experiments on frontier CUAs reveal significant vulnerabilities. Evaluation of four defense methods shows none provide adequate protection, with the best detection variant catching only 30% of injections.

**Reviewer Concerns:**

Reviewers raised several concerns: (1) limited novelty, as the hybrid sandbox builds upon existing frameworks and the benchmark uses cross-product expansion to reach 864 examples from a smaller core set. (2) Evaluation restricted to closed-source CUAs, with requests to include open-source models like UI-TARS and OpenCUA. (3) Superficial defense evaluation that tests existing methods rather than extending them. (4) Limited End2End evaluation scope (50 of 864 tasks) and lack of sensitivity analysis to UI noise. (5) Missing mechanistic analysis explaining why CUAs succumb to injection.

The authors provided thorough responses. They clarified that their key contribution is the integration enabling controlled hybrid web-OS adversarial testing and methodological features like Decoupled Eval for isolating robustness from capability. For agent selection, they explained that UI-TARS lacked public weights and OpenCUA lacked inference framework support, then added experiments with Claude 4.5 Sonnet and Qwen-3-VL. Defense evaluation was clarified as comprehensive across system-level  and model-level approaches, with results showing best detection at only 30%. The mechanistic analysis linked vulnerabilities to task-specific reasoning failures, supported by utility-under-attack results. A detailed failure taxonomy was provided addressing navigation, perception, and capability breakdowns.

While the sandbox builds on existing components, the authors demonstrate meaningful integration for hybrid adversarial testing unavailable elsewhere. The empirical findings are alarming and timely. The rebuttal addressed most concerns with new experiments on additional CUAs, failure taxonomies, and comprehensive defense characterization, though E2E evaluation scope and UI noise sensitivity remain limitations acknowledged by cost constraints.

**Reviewer Scores:**

y1z7:4->6 Explicitly raised score; agent selection, defense evaluation, and mechanistic analysis concerns adequately addressed.
MWPP: 8->8 Already positive; minor concerns about utility estimation and DoomArena comparison were addressed.
aku4: 6->6 Novelty and attack diversity concerns partially addressed; failure taxonomy addition is valuable but core concerns remain.
Djvv:6->6 7Open-source CUA concern addressed with new experiments; E2E scope limitation acknowledged but cost-justified.

---

### Decision · Program_Chairs · 2026-01-26

Accept (Oral)